# A Perspective on Cell Therapy and Cancer Vaccine in Biliary Tract Cancers (BTCs)

**DOI:** 10.3390/cancers12113404

**Published:** 2020-11-17

**Authors:** Shuting Han, Suat Ying Lee, Who-Whong Wang, Yu Bin Tan, Rachel Hui Zhen Sim, Rachael Cheong, Cherlyn Tan, Richard Hopkins, John Connolly, Wai Ho Shuen, Han Chong Toh

**Affiliations:** 1Division of Medical Oncology, National Cancer Centre Singapore, 11 Hospital Crescent, Singapore 169610, Singapore; han.shuting@singhealth.com.sg (S.H.); lee.suat.ying@singhealth.com.sg (S.Y.L.); nmowww@nccs.com.sg (W.-W.W.); Rachael.cheong.k.m@nccs.com.sg (R.C.); cyt26@cam.ac.uk (C.T.); timothy.shuen.w.h@nccs.com.sg (W.H.S.); 2Singapore Health Services, 31 Third Hospital Ave, #03-03 Bowyer Block C, Singapore 168753, Singapore; yubin.tan@mohh.com.sg (Y.B.T.); rachel.sim@mohh.com.sg (R.H.Z.S.); 3Institute of Molecular and Cell Biology (IMCB), A*STAR, 61 Biopolis Drive, Singapore 138673, Singapore; rhopkins@imcb.a-star.edu.sg (R.H.); jeconnolly@imcb.a-star.edu.sg (J.C.)

**Keywords:** biliary tract cancer, review, immunotherapy, cell therapy, personalized medicine, dendritic cell vaccine, bevacizumab

## Abstract

**Simple Summary:**

In this review, we discuss treatment strategies in biliary tract cancers (gallbladder cancer, intrahepatic cholangiocarcinoma and extrahepatic cholangiocarcinoma). In particular, we will describe advances in cellular therapies and cancer vaccines for biliary tract cancers, followed by our local experience with combining a melanoma-associated antigen (MAGE)-positive cell lysate-based autologous dendritic cell vaccine and anti-angiogenic therapy (bevacizumab) in a case of stage IV gallbladder cancer.

**Abstract:**

Biliary tract cancer (BTC) is a rare, but aggressive, disease that comprises of gallbladder carcinoma, intrahepatic cholangiocarcinoma and extrahepatic cholangiocarcinoma, with heterogeneous molecular profiles. Advanced disease has limited therapeutic options beyond first-line platinum-based chemotherapy. Immunotherapy has emerged as a viable option for many cancers with a similar unmet need. Therefore, we reviewed current understanding of the tumor immune microenvironment and recent advances in cellular immunotherapy and therapeutic cancer vaccines against BTC. We illustrated the efficacy of dendritic cell vaccination in one patient with advanced, chemorefractory, melanoma-associated antigen (MAGE)-positive gallbladder carcinoma, who was given multiple injections of an allogenic MAGE antigen-positive melanoma cell lysate (MCL)-based autologous dendritic cell vaccine combined with sequential anti-angiogenic therapy. This resulted in good radiological and tumor marker response and an overall survival of 3 years from diagnosis. We postulate the potential synergism of adding anti-angiogenic therapy, such as bevacizumab, to immunotherapy in BTC, as a rational scientific principle to positively modulate the tumor microenvironment to augment antitumor immunity.

## 1. Introduction

Biliary tract cancer (BTC) is a rare but clinically aggressive cancer that accounts for 3–5% of cancer diagnoses globally. It is a heterogeneous disease comprising gallbladder carcinoma (GBC), intrahepatic cholangiocarcinoma (IHCC) and extrahepatic cholangiocarcinoma (EHCC). With an increased understanding of the differing anatomical, histological and molecular subtypes of biliary tract cancers, novel therapies have emerged in the last two decades. However, chemotherapy remains the standard of care in first-line metastatic biliary tract cancers with modest efficacy. As such, more research efforts are required in the subsequent lines of treatment for patients with advanced stage disease.

## 2. Current Treatment Strategies in Advanced BTC

There is still a paucity of highly efficacious chemotherapy to treat advanced BTC. Following the landmark phase III trial ABC-02, combination gemcitabine and cisplatin found first-line indication in the treatment of advanced BTC, but with a median overall survival (OS) of only slightly under a year [1]. More recently, a Japanese phase III trial comparing gemcitabine-Tegafur/Gimeracil/Oteracil (TS-1^®^) with the established gemcitabine-cisplatin regimen proved non-inferiority of the aforementioned regimen [2]. Other internationally acceptable regimens include gemcitabine-oxaliplatin and gemcitabine-capecitabine, both supported by phase II data. Subsequently, dose intensification of gemcitabine-cisplatin with the addition of TS-1 [3] or nab-paclitaxel [4] has been attempted. While these increased the response rates and overall survival by a small margin, they also resulted in higher toxicities. Second-line options include fluoropyrimidine-based chemotherapy (e.g., FOLFOX) [5] in combination with oxaliplatin or irinotecan, but overall survival results remain disappointing.

The advent of molecular testing and next-generation sequencing (NGS) have unveiled potential treatment options for chemo-refractory disease. Biliary tract cancers have multiple molecular alterations and significant intra-tumoral heterogeneity, with variations of molecular profile across IHCC, EHCC and GBC. Main oncogenic mutation pathways include that of Wnt-Catenin Beta 1 (WNT-CTNNB1), MYC, ErbB, tumor necrosis factor (TNF) and vascular endothelial growth factor (VEGF) signaling, and this is part of a complex genomic network that modulates cell cycle regulation, DNA damage, MYC amplification, epigenetic regulation, demethylation, kinase signaling and immune dysregulation [6]. In the MOSCATO trial, 68% of patients with biliary tract cancer (*n* = 43) were identified to have potentially actionable mutations, 33% of which had an objective response [7]. Recently, a few targetable mutations have emerged as well.

The fibroblast growth factor (FGF) pathway upregulates the mitogen-activated protein kinase (MAPK) and phosphatidylinositol 3-kinase-A (PI3KA) pathways; FGF receptor 2 (FGFR2) gene alterations are particularly involved in the pathogenesis of cholangiocarcinoma (CCA) and harbored by about 9–16% of patients. In the FIGHT-202 trial [8], pemigatinib (selective, oral inhibitor of FGFR1, 2 and 3) was found to achieve high response rates (36%, including three complete responses) and disease control rates (80%) in patients with FGFR2 fusions or rearrangements as compared to those who had no FGF/FGFR alterations, with a duration of disease control of about 7.5 months. This led to an accelerated US Food and Drug Administration (FDA) approval. A phase III trial comparing pemigatinib to gemcitabine-cisplatin in the first-line setting for patients with advanced BTC with FGFR2 alterations is also underway.

Isocitrate dehydrogenase 1 (IDH1) is a key enzyme involved in the citric acid cycle, and mutations in this enzyme have been linked to oncogenesis. IDH1 mutations are present in 25% and IDH2 mutations in 3% of BTC patients, particularly in intrahepatic cholangiocarcinoma [9]. ClarIDHy is a phase III trial that explored Ivosidenib (small molecule inhibitor of IDH1) in patients with IDH-mutated BTC, and showed improved progression-free survival (PFS) and overall survival even in pre-treated patient populations [10].

Neurotrophic tyrosine receptor kinase (NTRK) rearrangements are rare in incidence in BTC (<5%), but could still be a potential drug target. Larotrectinib or Entrectinib are approved for patients with such rearrangements in view of durable response rates of up to 75% in a phase II study [11]. There has also been promising antitumor activity demonstrated in BRAF-mutated and epidermal growth factor receptor (EGFR)-mutated BTC patients, such as the use of Dabrafenib and Trametinib [12], as well as Erlotinib alone or in combination with Bevacizumab [13]. 

BTCs usually do not have high mutational burden; in a report of 239 BTC tumors, where whole exome sequencing (WES) was performed, the median numbers of mutations across the IHCC, EHCC and gallbladder cancer subtypes were 39, 35 and 64, respectively, with only 5 patients showing a deficient mismatch repair (dMMR) or microsatellite instability-high (MSI-H) genotype [14]. The MSI-H or dMMR status confers a higher neoantigen load and tumor mutational burden (TMB), leading to a potential increase in responsiveness to immune checkpoint inhibitors. The KEYNOTE 158 study highlighted how MSI-H BTC patients had response rates as high as 40% with duration of responses more than 2 years, and that patients with high TMB also correlated with greater benefit from immunotherapy [15]. Even for patients with microsatellite-stable (MSS) BTC, other biomarkers like programmed death ligand 1 (PD-L1) overexpression also predict for better response from immunotherapy, as evidenced by a phase II trial demonstrating partial response in patients with MSS BTC treated with nivolumab in the second-line setting [16]. 

Emerging data suggest that in a biomarker unselected population, there can also be durable responses to combination immunotherapy such as Nivolumab and Ipilimumab [17]. Gemcitabine-cisplatin was combined with durvalumab alone or in addition to Tremelimumab in advanced BTC, and showed promising efficacy and tolerability with response rates at 70% and disease control rates (DCR) in 90% of patients with good response durability [18]; this is currently being further evaluated in the phase III TOPAZ-1 trial. In addition, gemcitabine-cisplatin was also combined with nivolumab in another phase II trial [19] and similarly showed clinical efficacy with comparable DCR and manageable toxicities. 

## 3. Inflammation and BTC 

The causative link between inflammation and biliary tract cancer has been well established in various types of BTC—the association of gallbladder carcinoma with cholelithiasis-induced chronic inflammation, EHCC with parasites (e.g., liver fluke infections) and chronic pancreatitis and IHCC with persistent hepatolithiasis and hepatitis viruses [20,21,22]. Primary biliary sclerosis and primary sclerosing cholangitis are autoimmune conditions that also lead to inflammation and increased risk for BTC development [23]. 

One study suggested that chronic inflammation may lead to biliary tract cancers through aberrant activation-induced cytidine deaminase (AID) expression, resulting in the generation of somatic mutations in key cancer genes including TP53, c-MYC and promotor region of INK4A/p16. Inflammatory cytokines such as tumor necrosis factor-alpha (TNF-α), transforming growth factor-beta (TGF-β), interleukin-6 (IL-6) and other cytokines/chemokines have been known to enhance tumor proliferation, activate epithelial-to-mesenchymal transition (EMT) and promote metastatic potential [24,25,26]. In particular, sustained IL-6-STAT3 signaling was shown to contribute to tumorigenesis and cholangiocyte transformation during chronic inflammation or fibrosis via the alteration of EGFR promotor methylation and inhibition of apoptosis [27]. Furthermore, some small studies have shown correlation between IL-6 and overall survival outcomes—patients who had a higher level of IL-6 levels tended to have a higher systemic burden and also worse overall survival [28]. More studies are required to evaluate the role of IL-6 as not only a potential target, but also a biomarker in BTC. 

Notably, cholangiocarcinomas are often found to harbor both inflammatory cells (T cells, B cells, natural killer cells, neutrophils and macrophages) and a high amount of tumor stroma, including vasculature, extracellular matrix (ECM) and fibroblasts including cancer-associated fibroblasts (CAFs) and mesenchymal stromal cells (MSCs) [29,30]. Key growth factors such as platelet-derived growth factor (PDGF-DD), FGF and TGF-β also aid CAF infiltration and other mediators (some in the form of exosomes) influence crosstalk with CCAs [31]. One report analyzed the TME in over 500 IHCC and found that while 45% were immune-desert (“cold” phenotype), more than 50% of CCAs were immune-infiltrated with 11% showing very high immune cell infiltration (“hot” phenotype) [32]. It was also shown that the tumor stroma in CCA is composed of α-smooth muscle actin (α-SMA)-positive CAFs that outweigh the tumor itself and form a barrier against the infiltration of immune cells [30,33]. Furthermore, transcriptomic results from 10 IHCC tumors found high vascular endothelial growth factor (VEGF) expression, which is thought to induce remodeling of the tumor microenvironment (TME) and limit T-cell infiltration, therefore leading to lower responsiveness to immunotherapy [34]. Immunosuppressive mediators such as indoleamine 2,3-dioxygenase (IDO), interleukin-10 (IL-10) and TGF-β are also highly expressed in CCAs; these mediators can suppress the function of effector, cytotoxic T lymphocytes (CTL) and increase the migration of innate immune cells such as tumor-associated neutrophils (TANs), tumor-associated macrophages (TAMs) and myeloid-derived suppressor cells (MDSCs) into the tumor [35].

Apart from an immunosuppressive TME, studies also revealed immune escape mechanisms of certain BTCs in the form of major histocompatibility complex-I (MHC-I) downregulation, which was associated with poorer prognosis in BTC and lower tumor-infiltrating lymphocyte (TIL) infiltration [36]. In the same report, TIL infiltration was noted in about half of the 375 resected BTC tumors and it correlated positively with overall survival. Other studies also showed the abundance of CD8^+^ T cells within the tumor, while more CD4^+^ T cells were found in the tumor–liver interface [37], and an association between longer overall survival and the presence of tumor-infiltrating CD4^+^ or CD8^+^ T cells [38,39,40].

## 4. Cellular Immunotherapy and Cancer Vaccine in BTC

Though immune checkpoint inhibitors have therapeutically revolutionized the field of oncology, their efficacy in advanced BTC is still under active investigation. In view of the limited effective treatment options, there has been increased research in personalized cell-based therapy and cancer vaccines, either as a stand-alone treatment or in combination with other systemic therapies. Among cell therapy approaches, adoptive T-cell therapy, peptide-derived vaccines and dendritic cell (DC) vaccines have emerged with positive clinical outcomes in case reports and early phase clinical trials.

Adoptive T-cell therapy refers to the process where patients’ T lymphocytes are extracted and isolated from tumor or blood. They are then activated and expanded in vitro, some with genetic modification, then re-infused usually following lymphodepletion chemotherapy. Such therapies can exist in the form of autologous TIL infusion, chimeric antigen receptor T-cell (CAR-T) therapy targeting specific surface tumor antigens and genetically-modified (GM) T-cell receptor (TCR) T-cell therapy.

A case report in 2006 first suggested the clinical efficacy of adoptive cell therapy in BTC. A patient with IHCC with lymph node involvement received radical resection and was treated with adjuvant CD3-activated T cells plus tumor lysate- or peptide-pulsed dendritic cells and survived over 3.5 years following treatment [41]. Subsequently, Rosenberg’s team in National Institute of Health (NIH) reported in 2014, a case of a metastatic cholangiocarcinoma patient, who received TIL infusion that had been co-cultured with antigen-presenting cells (APCs) transfected with somatic non-synonymous mutations identified in the tumor, resulted in tumor regression for 7 months [42]. An adjuvant trial then reported 36 BTC patients post-resection, who received activated T-cell transfer and vaccinated with autologous tumor lysate-pulsed DCs. The median PFS was improved at 18.3 months in patients who received adjuvant immunotherapy compared with 7.7 months in those with surgery alone (*p* = 0.005). Overall survival reached 31.9 months in patients receiving adjuvant immunotherapy compared with 17.4 months (*p* = 0.022) in patients who did not receive it [43]. A recent phase I study of 11 human epidermal growth factor receptor 2 (HER2)-positive advanced BTC (*n* = 9) and pancreatic adenocarcinoma (*n* = 2) patients treated with HER2-targeted CAR-T therapy demonstrated disease control in 4/9 BTC patients, which included a partial response (PR) that lasted 4.5 months. The infusion of CAR-T-HER2 cells was safe without significant adverse events, except for one grade 3 (G3) febrile event and another G3 transaminitis [44]. 

As immunogenic targets, tumor-associated antigens (TAAs) such as Wilms’ tumor protein 1 (WT1) and mucin protein 1 (MUC1) emerged as potential peptide vaccine options in BTC. WT1 mutations were found in about 80% of BTC, while MUC1 was reported to be overexpressed in 90% of BTC. Both are associated with worse prognosis across various cancers [45]. A dose-escalation phase I study of WT1 peptide vaccination in combination with gemcitabine in 8 patients with advanced BTC reported stable disease at 2 months in 4/8 patients and a median OS of 288 days. WT1-specific T cells were detectable in 5/8 of patients after vaccination [46]. Another phase I trial of MUC1 peptide vaccine was conducted in 8 patients with advanced pancreatic and biliary cancers with good safety profile, though disease progression was noted in 7/8 patients [47].

Subsequently, Human Leukocyte Antigen (HLA)-matched multiple peptide vaccinations were attempted to increase specific antitumor responses. In one case report, whole exome sequencing, whole transcriptomic sequencing and HLA ligandome analyses were used to identify the immunopeptidome that led to the creation of a personalized multi-peptide vaccine (7 non-mutated tumor-associated peptides) emulsified in adjuvant Montanide. This vaccine was administered as multiple dosing to a 56-year-old patient with advanced IHCC, who had also undergone several surgeries for removal of the primary tumor, locoregional recurrences and lung metastasis. She remained recurrence-free for over 41 months [48]. A phase I trial reported a quadruple peptide vaccine therapy (HLA-A*2402-restricted epitope peptides) in 9 patients with advanced and chemorefractory BTC [49]. Peptide-specific T-cell immune responses were observed in 7/9 patients and clinical responses were observed in 6/9 patients. The median PFS and OS were 156 and 380 days, respectively. In a subsequent phase I trial by the same authors, a 3-peptide combination vaccine that includes cycle association protein 1 (CDC1), cadherin 3 (CDH3) and kinesin family member 20A (KIF20A) was administered to 9 patients with advanced BTC [50]. Again, this was well tolerated and resulted in peptide-specific T-cell responses in all patients, with stable disease seen in 5/9 patients. A separate phase II multi-peptide vaccine trial was reported, with the difference of each vaccine being personalized, selecting up to four HLA-matched peptides. This vaccine induced T-cell responses in 8/17 patients who completed the first cycle of vaccination and 4/7 patients following the second cycle [51]. Personalized multi-peptide vaccination remains labor-intensive, but appears to have better tumor control in selected BTC patients.

Interestingly, a phase II trial reported the combined use of Gemcitabine with Elpamotide, an HLA-A*24:02-restricted epitope peptide of VEGF receptor (VEGFR)-2. The reported response rate was 18.5%, with median survival of 10.1 months, which was longer than that of the historical control, and the 1-year survival rate was 44.4%. Injection site reactions were observed in 64.8% of these patients, suggesting the presence of delayed hypersensitivity caused by sensitization of peptide-reactive T cells. In these patients, median OS was significantly longer (14.8 months) compared to those with no injection site reactions (5.7 months) [52].

Dendritic cells are potent professional APCs and effective therapeutic cancer vaccine vehicles that present tumor antigens. The only therapeutic cancer vaccine that has been approved by the FDA is Sipuleucel-T, an autologous active cellular immunotherapy that consists of activated monocytes as APCs presenting a recombinant fusion protein of prostatic acid phosphatase fused to immunostimulant granulocyte-macrophage colony-stimulating factor (GM-CSF) against metastatic castrate-resistant prostate cancer, demonstrating improvement in overall survival [53]. In one retrospective study, WT1 and MUC1-pulsed DC vaccination was given to 65 patients with unresectable or recurrent BTC with good tolerability. The objective clinical response remained modest with partial response in only 6% of patients, stable disease in 23% of patients and median survival of 7.2 months from the time of vaccination [54]. Another phase I/II study also used MUC1-loaded DC vaccine as adjuvant therapy in 12 patients with resected BTC and pancreatic cancer. Despite the detection of increased CD8^+^ and CD4^+^ T-cell responses, vaccination did not induce anti-MUC1 antibody responses. Median OS was 26 months with 33% of patients without disease recurrence after 4 years [55]. These results again show the potential efficacy of tumor-associated antigen (specifically WT1 and MUC1)-loaded DC vaccines in disease control and it remains an active area of investigation. Tumor lysate-based DC vaccines in BTC remain as investigational therapies, though early efficacy was seen in in-vitro studies [56].

Strategies to improve the clinical efficacy of a therapeutic cancer vaccine or adoptive T-cell therapy in BTC include discovery of personalized neoantigens to tailor greater personalized immunotherapy. Preclinical organoid or cell line models may be able to reveal key mutations and mediators that influence TIL infiltration or response to immunotherapy, and can potentially aid in future cell therapy development [57]. Separately, deep integrative genomic analysis can uncover significant and potentially immunogenic genetic mutations. For example, a recent large study on GBC uncovered significantly mutated genes related to ERBB2, ERBB3, KRAS, PIK3CA and BRAF, and neoantigens from several mutated GBC genes including ELF3, ERBB2 and TP53 that were proven to activate T cells, therefore being potential candidate antigens for a therapeutic cancer vaccine. In particular, frame shift mutations in ELF3 led to the identification of several immunogenic cancer neoantigens [58].

Carcinogenesis in BTC arises through a complex interaction between extracellular ligands and intracellular signaling pathways—an amalgamation of pro-inflammatory cytokines and growth factors as well as aberrant activation and deregulation that orchestrate disinhibited cell proliferation and genetic/epigenetic alterations. It is therefore important to further understand and hence better modulate the tumor immune microenvironment of the BTC to overcome the resistance mechanisms toward cell therapy. Overcoming the immunosuppressive TME through combination approaches such as with immune checkpoint inhibitors, targeted therapy and chemotherapy may potentially improve future treatment outcomes of cell therapy. 

## 5. A Case of Refractory GBC that Achieved Clinical Benefit with DC Vaccination in Combination with Anti-Angiogenic Therapies

Our institution, National Cancer Centre Singapore (NCCS), had previously reported a phase II clinical trial of an allogeneic melanoma cell lysate (MCL)-based autologous DC vaccine in 20 metastatic colorectal cancer patients, whose tumors were biopsy-proven for expression of melanoma-associated antigen (MAGE)-A [59]. This DC vaccine subsequently obtained NCCS’ Institutional Ethics Committee approval as part of a non-profit, named-patient compassionate use program for gastrointestinal cancer patients who progressed after conventional systemic treatments and were no longer recommended for any further standard therapies. Of the 78 real-world patients who were treated on the MCL-based DC vaccine compassionate use program, 7 received DC vaccine and bevacizumab (DC+Bev) combination and the most recently treated patient had advanced gallbladder cancer with confirmed MAGE-A expression. To date in 2020, this case remains as the only gallbladder cancer treated with the DC+Bev compassionate use program. 

Our patient presented in December 2013 with recurrent MSS gallbladder cancer with liver, lung and peritoneal metastases. He was 83 years old, had poor performance status (ECOG 2) and had received multiple lines of prior systemic chemotherapy with radiologically proven progression of disease before he was referred to NCCS. He started treatment with DC+Bev in December 2014 and achieved a dramatic reduction in Carbohydrate Antigen (CA) 19-9 and neutrophil-to-lymphocyte (NLR) ratio (Figure 1) after three cycles of treatment. Repeat positron emission tomography-computed tomography (PET-CT) scan in January 2015 showed significant PR with resolution of the 2.3 cm liver metastasis and near resolution of the lung metastasis (Figure 2). After an episode of enterocutaneous fistulation and gallbladder abscess, bevacizumab was discontinued and DC vaccination was resumed with oral thalidomide, which also has anti-angiogenic properties, but no known risks of fistulation. On this combination, his CA 19-9 levels declined and he remained with stable disease until the 15th cycle of DC vaccine. Following this, the patient received anti-programmed death 1 (PD1) immunotherapy (nivolumab) at a private oncology clinic with rapid disease progression. He was then re-challenged with a second course of DC + thalidomide combination. Significant clinical response was observed again, with CA 19-9 levels reducing by more than 80% from over 10,000 U/mL to 1251 U/mL. PET-CT scan in February 2016 again revealed PR in the gallbladder mass and peritoneal nodules. Eventually, he progressed on treatment and subsequently passed away in December 2016 after stopping all treatments in November. 

## 6. Correlative Studies

A MAGE-A gene expression analysis confirmed MAGE-A4 expression in his GBC tumor, which was one of the MAGE-A species found in the allogenic tumor cell lysate. A multiplex gene sequencing analysis (ACT Genomics, Singapore) found the cancer to contain KRAS and ABL1 mutations. PD-L1 staining by immunohistochemistry (IHC) showed weak expression. Immune profiling of the serially collected peripheral blood samples showed a shift during the period of treatment response compared to the pre-treatment baseline. Using the Luminex assay, it was observed that, while the pre-treatment plasma cytokine levels of IL-1a, IL-1b, IL-2, IL-6, IL-10, VEGF and FGF2 were all higher than baseline reference ranges for healthy individuals, they were found to decrease during periods of treatment response compared to pre-treatment and disease progression time points (Figure 3). Peripheral blood immune cell profiling by flow cytometry analysis showed that the frequency of monocytic myeloid-derived suppressor cell (MDSC, CD14^+^HLADR^−^) population corresponded with cancer progression and treatment—lower during periods of treatment response and higher during pre-treatment and disease progression. The overall CD4 and CD8 T-terminal effector (TEMRA) cells, on the other hand, decreased as the disease progressed (Figure 4). 

## 7. Discussions

Our patient had a good response during 2 years of combined DC vaccine and anti-angiogenic therapy, and survived 3 years from diagnosis of stage IV GBC with a good quality of life. It is noteworthy that this patient had previously received multiple systemic therapies including bevacizumab prior to consulting us at NCCS. We are encouraged by this finding and would like to postulate the potential synergism between anti-angiogenic therapy and the DC vaccine in selected patients, though this needs to be confirmed through larger clinical trials.

The addition of the anti-VEGF-A antibody, bevacizumab, to the DC vaccine therapy aims to harness a potential synergistic effect. The mode of action for bevacizumab has been described, including anti-angiogenesis, remodeling the immunosuppressive TME to favor effector T-cell trafficking, infiltration and optimal killing [60,61]. However, its single agent activity across cancers remains minimal, being most useful as part of a chemotherapy backbone or together with other biologic agents [62]. Given its pivotal role of normalizing tumor vasculature and reprogramming the tumor immune microenvironment, there is rationale for combining cancer immunotherapy with anti-angiogenic therapy to improve the treatment outcome. Anti-VEGF therapy had also been successfully combined with DC vaccine therapy in the context of advanced glioblastoma [63] and ovarian cancers [64], both with positive safety and clinical signals. We posit that combination anti-angiogenic therapy may have a strong role in improving cell-based therapy in advanced BTC, given the high amount of hypoxia/desmoplasia, VEGF expression and stromal CAFs in the tumor microenvironment of these cancers. 

We had previous clinical experience of using thalidomide combinations in advanced gastro-esophageal adenocarcinoma [65] and hepatocellular carcinoma [66]. Thalidomide is well documented to possess both anti-angiogenic and immunomodulatory properties [67,68]. The aforementioned case is the first clinical evidence combining DC vaccination and thalidomide in any solid tumor to our knowledge. Previously, there were only a few clinical reports of the use of the thalidomide analog lenalidomide as an immune adjuvant to DC vaccines in hematological malignancies [69,70]. In vivo findings from these studies confirmed lenalidomide’s immunomodulatory effects; vaccine-specific immune responses and critical maintenance of T-cell persistence in vivo were higher with its concomitant administration. Pre-clinical data demonstrated both antitumor cytotoxic effects as well as its effect as a vaccine immune modulator.

## 8. Conclusions

There remains an unmet need for BTC patients who are chemo-refractory or do not harbor specific targetable mutations. Encouraging clinical signals are seen in earlier clinical trials of therapeutic cancer vaccination and adoptive T-cell therapy in these patients, indicating a potential, emerging role for cellular immunotherapy against BTC. We believe that our real-world case of an elderly patient with heavily pre-treated GBC who responded to DC vaccination in combination with sequential anti-angiogenic drugs is an example of the nascent yet possible role of cancer vaccines in BTC. It also illustrates the importance of choosing rational combinations to overcome the immunosuppressive tumor microenvironment and augment an antitumor immune effect. The increasing utilization of personalized, tumor-specific antigens with next-generation sequencing and neoantigen predictive platforms harnessed as targets for immunotherapy is also highly anticipated.

## Figures and Tables

**Figure 1 cancers-12-03404-f001:**
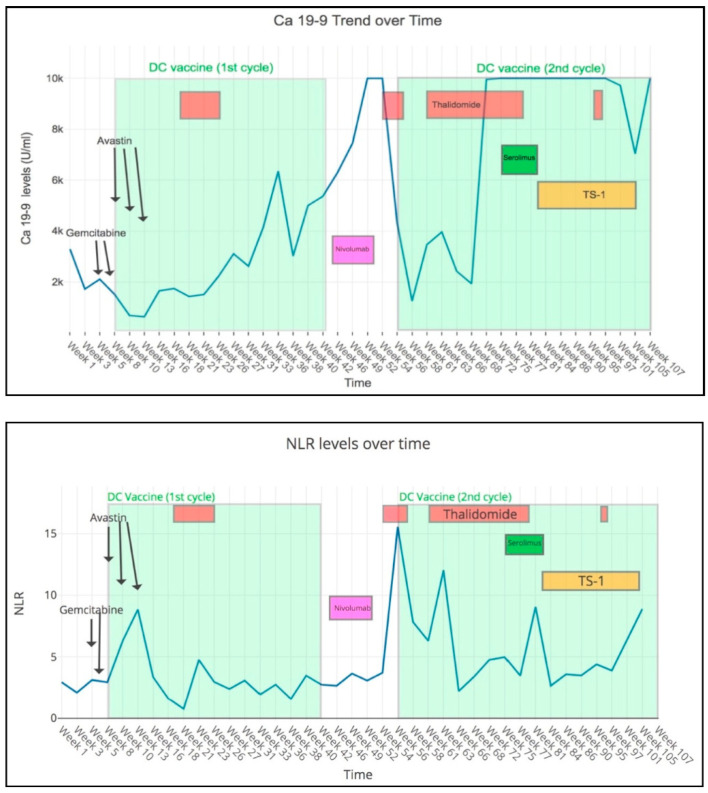
Carbohydrate Antigen (CA) 19-9 and neutrophil-to-lymphocyte ratio (NLR) of the patient during treatment.

**Figure 2 cancers-12-03404-f002:**
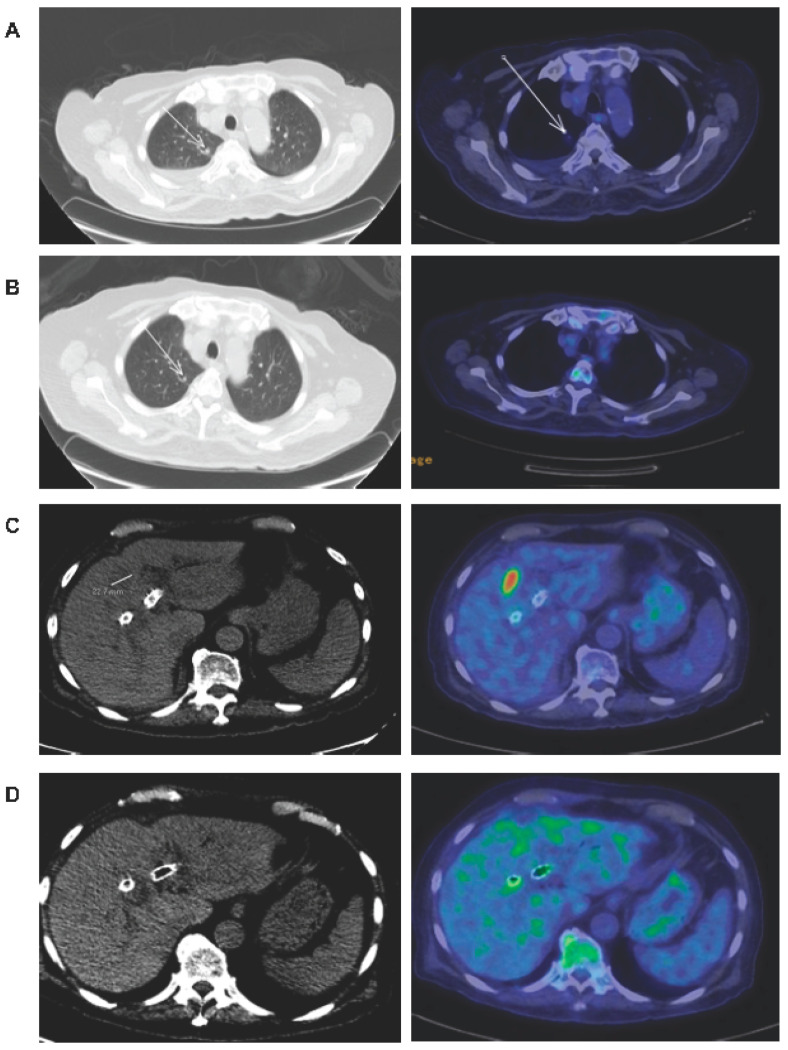
Resolution of liver and lung metastasis in the patient after commencing dendritic cell (DC) vaccine and bevacizumab. (**A**) Positron emission tomography-computed tomography (PET-CT) done prior to any treatment in week 3 showing 0.5 cm lung metastasis. (**B**) PET-CT done in week 13 showing a reduction in size of lung metastasis that became unmeasurable. (**C**) PET-CT done prior to any treatment in week 3 showing a 2.3 cm liver metastasis. (**D**) PET-CT done in week 13 showing resolution of the liver metastasis.

**Figure 3 cancers-12-03404-f003:**
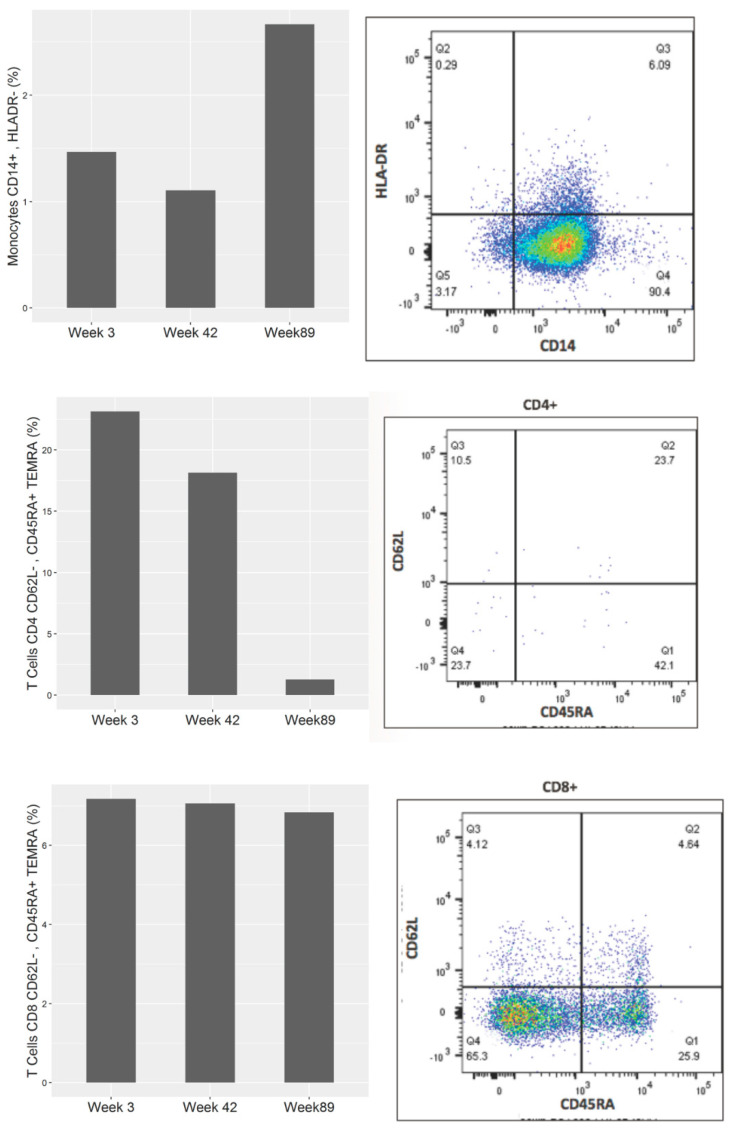
Flow cytometry analysis of peripheral blood mononuclear cells (PBMCs) to identify percentages of monocyte CD14^+^HLADR^−^ cells, representative of the monocytic myeloid-derived suppressor cell (MDSC) population, CD4 and CD8 T-effector cells prior to treatment (week 3) after the first cycle of DC vaccine + anti-angiogenic combination (week 42 and during disease progression (week 89).

**Figure 4 cancers-12-03404-f004:**
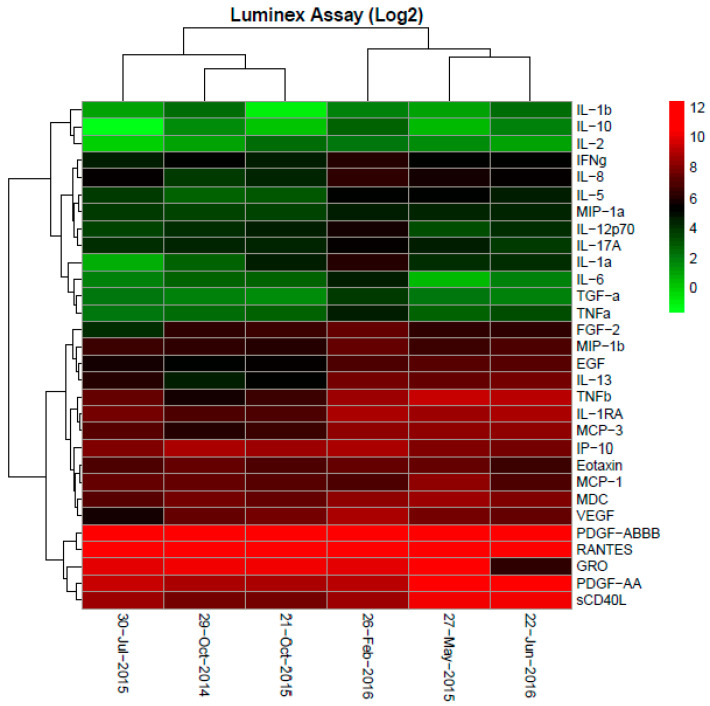
Plasma cytokine profile of Patient 7. Using the Luminex assay, it was observed that, while the plasma cytokine levels of interleukin (IL)-1a, IL-1b, IL-2, IL-6, IL-10, VEGF and fibroblast growth factor 2 (FGF2) were all higher than baseline reference ranges for healthy individuals pre-treatment [10]; they were found to be lowered during periods of treatment response compared to pre-treatment and disease progression time points.

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
