# Peer review of "A Perspective on Cell Therapy and Cancer Vaccine in Biliary Tract Cancers (BTCs)"

_cancers, 2020, doi:10.3390/cancers12113404_

Round 1

Reviewer 1 Report

The review by Han and colleagues discuss different treatment options in biliary tract cancers, including intrahepatic cholangiocarcinomas, extrahepatic cholangiocarcinomas, gallbladder cancers. Of note, the authors provide a comprehensive work reporting recent advances in cellular therapiesz and cancer vaccines in biliary tract cancers, also reporting an experience regarding a patient with melanoma-cell lysate based dendritic cell vaccine and bevacizumab.

The review is quite well written and organized.

Figures are clear.

In our opinion, the paper would deserve a linguistic revision by professional service.

The authors included papers of notable importance in this setting.

However, we suggest including the following references regarding experimental models in cholangiocarcinoma and immunotherapeutic approaches in this heterogenous group of aggressive malignancies:

doi: 10.3390/cancers12082308

doi: 10.21873/anticanres.14282.

We believe this article is suitable for publication in the journal although minor revisions are needed. The main strengths of this paper are that it addresses an interesting and timely question and provides a clear answer, with some limitations. We suggest including some references and performing a linguistic revision.

Author Response

The review by Han and colleagues discuss different treatment options in biliary tract cancers, including intrahepatic cholangiocarcinomas, extrahepatic cholangiocarcinomas, gallbladder cancers. Of note, the authors provide a comprehensive work reporting recent advances in cellular therapiesz and cancer vaccines in biliary tract cancers, also reporting an experience regarding a patient with melanoma-cell lysate based dendritic cell vaccine and bevacizumab.

The review is quite well written and organized.

Figures are clear.

Reply: Thank you for your kind comments.

In our opinion, the paper would deserve a linguistic revision by professional service.

Reply: Thank you for your suggestions. We have done a thorough internal linguistic revision.

The authors included papers of notable importance in this setting.

However, we suggest including the following references regarding experimental models in cholangiocarcinoma and immunotherapeutic approaches in this heterogenous group of aggressive malignancies:

doi: 10.3390/cancers12082308

doi: 10.21873/anticanres.14282.

Reply: thank you and we have included the experimental model in CCA paper into our review.

We believe this article is suitable for publication in the journal although minor revisions are needed. The main strengths of this paper are that it addresses an interesting and timely question and provides a clear answer, with some limitations. We suggest including some references and performing a linguistic revision.

Reply: Thank you and have we have done some changes as suggested. please refer to our manuscript cover letter 2 for the detailed changes.

Reviewer 2 Report

A perspective on personalized cell therapies in 2 biliary tract cancers (BTCs) by Han et al. reviewed the cellular therapies and cancer vaccines as recent treatment options for biliary tract cancers. As there are limited treatment strategies for advanced BTCs, the possibility of such novel treatment options would be important in near future and would lead to the readers’ interest. I have some recommendations as following.

  1. As any cellular therapies are not common among clinicians in this field so far, I recommend that some basic information about cellular immunotherapy itself are summarized before the section 4 “Cellular immunotherapy in BTC”. It might include the classification of immunotherapy, the molecular mechanisms, the history of establishment of the treatment and some famous previous examples in cancers of other organs.
  2. Although this is a review article, a case report and the authors’ research data have a large proportion of the total manuscript. If applicable, other examples on cellular therapies from previous reports should be widely introduced, comparing the authors’ own experiences.
  3. As a background of immunotherapy, the relationship between inflammation and BTC is so important. In the section 3, authors described the linkage between inflammation and BTC, In the first paragraph of this section, authors only mentioned to the clinical relationship of them. There have been several reports describing the molecular mechanisms concerning the inflammation-associated carcinogenesis including BTC, such as Komori J et al “Activation-induced cytidine deaminase links bile duct inflammation to human cholangiocarcinoma” Hepatology 2008 or Maemura K et al “Molecular mechanism of cholangiocarcinoma carcinogenesis” J Hepatobiliary Pancreat Sci 2014 and others. Referring to these reports could improve the first paragraph of section 3 of this article.

Author Response

A perspective on personalized cell therapies in 2 biliary tract cancers (BTCs) by Han et al. reviewed the cellular therapies and cancer vaccines as recent treatment options for biliary tract cancers. As there are limited treatment strategies for advanced BTCs, the possibility of such novel treatment options would be important in near future and would lead to the readers’ interest. I have some recommendations as following.

Reply: thank you for your kind comments.

1. As any cellular therapies are not common among clinicians in this field so far, I recommend that some basic information about cellular immunotherapy itself are summarized before the section 4 “Cellular immunotherapy in BTC”. It might include the classification of immunotherapy, the molecular mechanisms, the history of establishment of the treatment and some famous previous examples in cancers of other organs.

Reply: thanks for the constructive suggestion. We have added an introduction section to immunotherapy, cell therapy and cancer vaccine for clinicians in this field. 

2. Although this is a review article, a case report and the authors’ research data have a large proportion of the total manuscript. If applicable, other examples on cellular therapies from previous reports should be widely introduced, comparing the authors’ own experiences.

Reply: Thank you. We have introduced a wide range of cell therapy and cancer vaccines, and have shortened our case report. There was also a discussion section describing the scientific rationale of our combination therapy. 

3. As a background of immunotherapy, the relationship between inflammation and BTC is so important. In the section 3, authors described the linkage between inflammation and BTC, In the first paragraph of this section, authors only mentioned to the clinical relationship of them. There have been several reports describing the molecular mechanisms concerning the inflammation-associated carcinogenesis including BTC, such as Komori J et al “Activation-induced cytidine deaminase links bile duct inflammation to human cholangiocarcinoma” Hepatology 2008 or Maemura K et al “Molecular mechanism of cholangiocarcinoma carcinogenesis” J Hepatobiliary Pancreat Sci 2014 and others. Referring to these reports could improve the first paragraph of section 3 of this article.

Reply: We have included the above suggested citations along with other relevant papers, and expanded upon the section on the link between chronic inflammation and carcinogenesis. Thank you!

Round 2

Reviewer 2 Report

The manuscript has been properly revised according to the reviewers’ comments.